Particulate matter resuspension from simulated urban green floors using a wind tunnel-mounted closed chamber

Seo Inhye 1
Park Chan Ryul 2
Yoo Gayoung gayoo@khu.ac.kr 1 3
1 Department of Applied Environmental Science, Kyung Hee University , Yongin , Republic of Korea
2 Urban Forests Division, National Institute of Forest Science , Seoul , Republic of Korea
3 Department of Environmental Science and Engineering, Kyung Hee University , Yongin , Republic of Korea
Wang Xiukang
Electronic publication date: 2023 Feb 8
Publication date: 2023
Volume: 11
Electronic Location ID: e14674
Received 2022 Aug 17; Accepted 2022 Dec 11
Copyright: ©2023 Seo et al.
Copyright year: 2023
Copyright holder: Seo et al.
License: This is an open access article distributed under the terms of the Creative Commons Attribution License, which permits unrestricted use, distribution, reproduction and adaptation in any medium and for any purpose provided that it is properly attributed. For attribution, the original author(s), title, publication source (PeerJ) and either DOI or URL of the article must be cited.
License URL: https://creativecommons.org/licenses/by/4.0/

Keywords: Urban green floor, Floor structure, Urban litter layer, Urban ecosystem service, Particulate matter, Resuspension, Management

Funding: Korea Ministry of Environment (MOE) 2022003560006 National Institute of Forest Science of Korea NIFOS FE0100-2019-03 This research was supported by the Korea Environment Industry & Technology Institute (KEITI) through “Climate Change R&D Project for New Climate Regime.”, funded by Korea Ministry of Environment (MOE) (2022003560006), and also supported by the National Institute of Forest Science of Korea (Grant No. NIFOS FE0100-2019-03). There was no additional external funding received for this study. The funders had no role in study design, data collection and analysis, decision to publish, or preparation of the manuscript.

==============================
Background

Green areas are thought to reduce particulate matter (PM) concentrations in urban environments. Plants are the key to PM reduction via various mechanisms, although most mechanisms do not lead to the complete removal of PM. Ultimately, PM falls into the soil via wind and rainfall. However, the fallen PM can re-entrain the atmosphere, which can affect plants capacity to reduce PM. In this study, we simulated an urban green floor and measured the resuspension of PM from the surface using a new experimental system, a wind tunnel-mounted closed chamber.

Methods

The developed system is capable of quantifying the resuspension rate at the millimeter scale, which is measured by using the 1 mm node chain. This is adequate for simulating in situ green floors, including fallen branches and leaves. This addressed limitations from previous studies which focused on micrometer-scale surfaces. In this study, the surfaces consisted of three types: bare sand soil, broadleaves, and coniferous leaves. The resuspended PM was measured using a light-scattering dust detector.

Results

The resuspension rate was highest of 14.45×10−4 s−1 on broad-leaved surfaces and lowest on coniferous surfaces of 5.35×10−4 s−1 (p < 0.05) and was not proportional to the millimeter-scale surface roughness measured by the roller chain method. This might be due to the lower roughness density of the broad-leaved surface, which can cause more turbulence for PM resuspension. Moreover, the size distribution of the resuspended PM indicated that the particles tended to agglomerate at 2.5 µm after resuspension.

Conclusion

Our findings suggest that the management of fallen leaves on the urban green floor is important in controlling PM concentrations and that the coniferous floor is more effective than the broadleaved floor in reducing PM resuspension. Future studies using the new system can be expanded to derive PM management strategies by diversifying the PM types, surfaces, and atmospheric conditions.

Introduction

Urban green spaces have been known to mitigate high atmospheric particulate matter (PM) concentrations, and their PM mitigation effect has been reported by previous studies (Deng et al., 2019; Heo & Bell, 2019; Jennings, Reid & Fuller, 2021). Gómez-Moreno et al. (2019) and Cohen, Potchter & Schnell (2014) measured PM concentrations at different distances from urban parks in Madrid (Spain) and Tel-Aviv (Israel), and they observed a lower concentration inside the park than outside, with a degree of 50% and 70% respectively. Yoo et al. (2020) measured the extent of PM reduction by passing through the urban park in Sihwa (Korea), and the reduction rates were up to 56.4%. Further, a number of researchers have reported that PM reduction occurs even in narrow green spaces, such as vegetation barriers in urban streets (Chen et al., 2021; He, Qiu & Pott, 2020; Kończak et al., 2021; Mori et al., 2018).

The main mechanisms of PM reduction by plants are adsorption and absorption (Shi et al., 2017; Wang, Zhao & Yao, 2019). While adsorption is the physical contact of PM with the plant surfaces such as trichomes, epicuticular wax, and bark, the absorption mechanism involves the entrance of PM to the stomata and cuticles (Escobedo & Nowak, 2009; Chen et al., 2017; Ha et al., 2021). When PM is absorbed through the plant’s stomata it is converted to less or non-toxic matter and stored in the vacuole or cell wall. This is the exhaustive removal process of PM although the amount is minor (McDonald et al., 2007; Wróblewska & Jeong, 2021). However, most PM is deposited and adsorbed on the vegetation surface and they enter the litter and soil layers or re-entrain into the atmosphere (Cai, Xin & Yu, 2019; Diener & Mudu, 2021; Henry, 2018; Xu et al., 2020). This is called PM resuspension, which is the mechanism by which PM deposited on vegetation and soil surfaces is removed from the surface by external factors such as wind and surface vibration and re-entrained back into the airflow (McPherson, Scott & Simpson, 1998; Ould-dada & Baghini, 2001; Henry & Minier, 2014).

Generally, PM resuspension is described as a function of the adhesion forces between the particle, the surface, and the fluid friction velocity (Loosmore, 2003; Nasr et al., 2019). The surface roughness is regarded as one of the most influential variables affecting PM resuspension because it has a significant impact on the adhesion forces on multi-scales (Qian, Peccia & Ferro, 2014). The largest scale of surface roughness that can affect PM resuspension could be the macrostructure of urban areas, such as buildings and tree canopies, and there are smaller surface roughness scales contributed by the shrubs/grass level (Salizzoni et al., 2008; Nasr et al., 2020). However, most of the studies have been conducted on microscale surface roughness such as leaf morphology using a single leaf (Speak et al., 2012; Wang, Shi & Wang, 2015; Chen et al., 2017; Xu et al., 2020). These studies focused on investigating the effect of leaf microstructure or chemical traits on PM resuspension and reported that the high roughness of leaf adaxial and abaxial surfaces contributed to PM capture (Zhang, Wang & Niu, 2017; Zhao et al., 2019).

Our question was whether these microscale properties of surface roughness of a single leaf could explain the effect of urban green floor on PM resuspension (Amato et al., 2013; Han et al., 2020). When focusing on the role of urban greenery in PM reduction, the scale to be noted should be expanded to a larger structure consisting of fallen branches, roots, leaves, and soil. At this scale, the physical structure of litter layer create boundary layer to make turbulence and the high surface roughness could lead to rather higher PM resuspension (Krogstad & Antonia, 1999; Volino, Schultz & Flack, 2009; Wu, Davidson & Russell, 1992). Despite of this complicated scale matter in PM resuspension, it is hard to find the studies on PM resuspension at the green floor level.

One of the main reasons why there are few previous studies simulating the effect of different urban green floor conditions on PM resuspension could be the absence of a proper experimental setup at this scale. The existing PM resuspension experiment settings are usually equipped with sophisticated controls over environmental conditions and contamination (Kim et al., 2016; Kottapalli & Novosselov, 2021; Li et al., 2022). As these experiments focus on the specific traits of the surface affecting resuspension, the scale of target surface is generally small (microscopic, nanoscopic) and only the pure surface materials were used. This small scale experimental setup is definitely not suitable for investigating the effects of green floor where the diverse litters and soil are mixed. Therefore, it is necessary to develop the proper experimental settings at the litter layer scale, which can function as the bridge between microscale pure surface experiment and in situ measurement of PM resuspension.

Using the developed experimental system, we examined the difference in PM resuspension rates by the urban green floor types with different surfaces, composed of bare soil, bare soil+coniferous litter, and bare soil+broadleaf litter. The working hypotheses were as follows:

(1) PM resuspension would be lower in the presence of litter layers compared to the bare soil due to higher surface roughness.

(2) PM resuspension would be different depending on whether it is a single leaf or a layer of leaves.

Materials & Methods

Resuspension chamber system design

We developed an acrylic chamber system (810 × 810 × 1,400 mm (L ×W ×H); Fig. 1) comprising a wind tunnel and a measurement chamber surrounding the tunnel. The wind tunnel has an inlet of PM (⌀ 10 mm) on the top and a target surface frame (300 × 100 mm (L ×W), 3 mm thick) on the bottom, where various types of surfaces can be placed. Detachable sliding doors are installed on both sides of the wind tunnel and a circulation fan (⌀ 80 mm) is in the center of the top side for scattering PM. The wind tunnel was mounted on a frame in the middle of the measurement chamber, and the main fan was placed on one side of the frame. The main fan was connected to an external controller that controlled the wind speed. The controller was programmed to set the wind speed at the desired intensity and duration up to 15 m s−1 in units of 1 m s−1.

Figure 1 Isometric views of the experimental settings and progress used in this study.

(A) The wind tunnel, (B) the test chamber with wind tunnel.

The measurement chamber has a rubber-packed door in front, where the wind tunnel slides horizontally. At each apex of the chamber, fans (⌀ 80 mm) were mounted for the internal circulation. To prevent experimental errors, antistatic films were attached to all the acrylic surfaces.

To verify whether the chamber system operated with the intended design, a verification experiment was conducted. The experiment aimed to verify the following: (1) the air only escapes through the outlet connected to the detecting sensor and not through the seams or door of the chamber; and (2) the air flow generated by the main fan and the mixing fans is well distributed throughout the chamber. To observe the air flow in the chamber, dry ice was placed in the center of the wind tunnel, and the sublimated gas from the dry ice was visualized by video.

As we intended to detect the total amount of PM scattered in the wind tunnel and that floating in the chamber, we connected the measurement device through the bottom hole (air outlet), and the device detected the PM concentrations by pumping air through the tubing. The measurement device was EDM 164 (Grimm, Baden-Wuerttemberg, Germany), which is a light-scattering dust detector that is widely used to measure atmospheric PM concentrations (Dahari et al., 2020; Münch et al., 2020; Kabelitz et al., 2021). Table 1 presents the performance evaluation results and details of EDM 164.

PM resuspension experimental design

Particulate matter sample

Standard reference material (SRM) No. 2786 of the National Institute of Standards & Technology (NIST) was used as the PM sample. SRM components included polycyclic aromatic hydrocarbons (PAHs), nitro-substituted PAHs, sugars, inorganic constituents and other natural atmospheric substances. This was in line with our experimental purpose of simulating the resuspension of atmospheric pollutants.

Information on the mass fractions and particle size distribution of the sample was provided by NIST and were measured using laser diffraction instruments (Mastersizer 2000; Malvern Panalytical Ltd, Malvern, UK; Table 2). The diameter of all the particles was less than 20 µm, and the mean particle diameter of the sample was less than 4 µm.

Table 1 Technical specification data of EDM164.

Parameter	Specification	
Particle size range	0.25–32 µm (31 size channels)	
Particle number	0–3,000,000 particles L−1	
Dust mass	0–100,000 µg m−3 (unit: 0.1 µg m−3)	
Reproducibility	>97% of total measuring range	
Accuracy	87.3%*	
Optical cell	Diode laser 660 nm	
Time resolution	6 s, maximum 60 min	
Sample flow rate	1.2 L min−1, ± 3% constant due to self-regulation	
Rinsing air	0.4 L min−1, protection of laser optics, reference air for self-test	
Sampling inlet	Heated, constant above ambient temperature	
Temperature range	−20–60 °C (−4–140 °F)	
RH range	<95%, non-condensing	
Notes.

Grimm Aerosol, Transportable Air Quality Monitoring Station EDM 164, https://www.grimm-aerosol.com/products-en/dust-monitors/mobile-pm-monitor/edm164/; Grimm, Baden-Wuerttemberg, Germany.

* EDM 164 Performance Certificate, https://www.grimm-aerosol.com/products-en/dust-monitors/mobile-pm-monitor/edm164/; Korea Conformity Laboratories, Incheon, South Korea.

Table 2 Particle size distribution information of NIST SRM No. 2786 sample.

	Size (µm)	
Mean particle diameter, d(0.5)	2.8	
Particle diameter, d(0.1)	0.91	
Particle diameter, d(0.9)	6.9	
Volume weighted mean	3.6	
Notes.

Certificate of Analysis Standard Reference Material® 2786; https://www-s.nist.gov/srmors/certificates/2786.pdf.

Target surface preparation

To simulate resuspension from different urban green floors, the target surfaces were prepared in three types: sand only, sand with broadleaf litter, and sand with coniferous litter. The sand-only surface consisted of a soil layer with a depth of 10 mm, and the other two surfaces consisted of a 10 mm soil layer and a litter layer (Fig. 2). The soil layer was composed of river sand to simulate the soil texture of urban soil (Jim & Ng, 2018; Caplan et al., 2019; Romzaykina et al., 2021). Fallen leaves were collected from an urban forest located in Suwon, Gyeonggi-do, Korea, and separated into broad and coniferous leaves. The species of the collected leaves were Quercus mongolica, Robinia pseudoacacia, and Quercus acutissima for broadleaves, and Pinus densiflora for coniferous leaves, which are commonly found in temperate forests. All materials were washed under running water and dried at 105 °C for more than 6 h.

Figure 2 Photograph of surfaces simulated in this study, whose frame was made of three mm thick fomex for easy transport.

(A) Sand surface. (B) Coniferous litter layer on the sand surface. (C) Broadleaved litter layer on the sand surface.

Experimental condition and chamber operation

A preliminary experiment was conducted to determine the optimal wind speed for the experiment and the amount of PM to be injected. Although all the soil and leaves were cleaned before the experiment to prevent PM suspension from the soil and leaves themselves, the background suspension from surface materials needed to be determined to eliminate data contamination. Thus, we ran the wind tunnel with different surfaces installed (four replications) without adding SRM 2786 and applied wind speeds of 1, 2, and 4 m s−1, which were in the range of the wind profile in the urban area; thus, the background suspension rate was identified (Memon, Leung & Liu, 2010; Chow et al., 2016). The results showed that significant differences in the amount of PM suspended by different surface types were evident only when the wind speed was 4 m s−1. Hence, we regarded a wind speed of 4 m s−1 as the optimum condition to differentiate the effect of the surfaces on the suspension. We also found that the average amount of background suspension was 5 mg per surface with 8 mg of maximum for all types of surfaces. Therefore, 9 mg of SRM material per surface was added, which is a sufficient amount of PM input to differentiate the signal from the background noise.

To reduce errors caused by environmental conditions, the air temperature and relative humidity of the chamber were continuously monitored using a digital climate measuring instrument (Model 480; Testo AG, Lenzkirch, Germany). The temperature and relative humidity inside of the chamber were maintained at 20.75–22.14 °C and 58.85–63.70%, respectively.

To deposit PM on each surface, a wind tunnel was closed with sliding doors on both sides, and PM was introduced into the top injection port of the tunnel (Fig. 1). The injected PM was dispersed by running the fan for 30 min and allowed to settle for at least one day (Qian, Ferro & Fowler, 2008). After the deposition was completed, the sliding doors were removed from both ends of the wind tunnel and immediately mounted on the frame inside the chamber. After the wind tunnel was mounted on the frame, the door of the chamber was closed and the concentration of PM inside the chamber was monitored. We stabilized PM using a small air purifier (CADR 13 m3 h−1, Model AP130MWKA, LG Electronics, KR) until the PM concentration in the chamber reached zero. The stabilizing procedure eliminated the influence of the external PM flowing into the chamber when the wind tunnel was mounted. If the PM concentration was kept low, the air purifier was stopped, and the main fan was operated for 10 min to resuspend the PM deposited on the target surface (Braaten, Paw & Shaw, 1990). We expected that once the main fan was on, the PM on the surface would be resuspended and blown on both sides of the wind tunnel. The suspended PM circulated inside the chamber and was finally collected through the hole at the bottom of the chamber connected with EDM 164. Figure 3 shows that the internal PM concentration increased due to resuspension of the PM when wind was applied; when the wind stopped, the concentration started to decrease and converged on zero over time. The time spent for each experimental run varied from 40 to 200 min, depending on how fast the PM settled. All experiments were repeated four times.

Figure 3 Raw data pattern detected by EDM164.

The x-axis was time (s) and the y-axis was PM concentration (µg m−3). The main fan was operated for 10 min (vertical dashed line on the left), and the concentration converged to zero after about 30 min (vertical dashed line on the right).

Measurements

Surface roughness

The roughness of the three surface types was measured using the roller chain method (Saleh, 1993). This method is easy, cost-efficient, and is widely used in field investigations (Gilley & Kottwitz, 1996; Jester & Klik, 2005; Thomsen et al., 2015). To measure the surface roughness in millimeters, a roller chain with a node length of one mm and a total length of 30 cm was used. The roller chain was placed diagonally on the target surface, and the length of the Euclidean distance between the two ends of the chain was measured. The measurements were conducted three times before scattering the PM on the surface. The result of the measurement was derived from the chain roughness (Cr) index and was determined using the following equation (Saleh, 1993): (1) Cr=1−L2/L1×100

where L1 is the distance over the surface (m) and L2 is the Euclidean distance (m).

To validate the roller chain method, we used stereo photograph image analysis, which is frequently used to measure soil surface roughness (Jester & Klik, 2005; Thomsen et al., 2015). The target surface was photographed at two points with the same height of 80 cm using a digital camera (D850; Nikon, Tokyo, Japan) with a 30 mm lens. The collected photographs were analyzed using ImageJ software (Schneider, Rasband & Eliceiri, 2012) and the roughness calculation plugin (Chinga et al., 2007). To compare the two indices, both were normalized by scaling to a range. Using the plot profile function of ImageJ, we also calculated the roughness density (λ) which is described as the following equation (MacDonald et al., 2016): (2) Λ= ∑kn/λ

where kn is the roughness height of each projection, and λ is the length of the surface without projections. The roughness density stands for the degree to which roughness projections are distributed per unit area or section.

PM resuspension

The concentration and particle size distribution of the resuspended PM were continuously measured at 6 s intervals using an EDM 164 placed at the bottom of the chamber. The amount and rate of resuspended PM were derived using the following equations: (3) ReMmg= ∑0tCn×Q×te

(4) ReRs−1=ReM/Mi×te

where ReM is the amount of resuspended PM (mg), t is the time taken from the start of the experiment until the concentration of PM in the chamber reaches zero (min), C is the concentration of PM in the chamber (µg m−3), Q is the inflow rate (L min−1), te is the operating time of the main fan (min), ReR is the rate of resuspended PM (s−1), and Mi is the initial input amount of PM (mg).

The particle size distribution of the resuspended PM sample was compared to that of the original PM sample. We modified the particle size distribution data provided by NIST using atmospheric particle density data and used it as the distribution of the original PM sample (Kim, Kim & Hwang, 2008). The total number of resuspended particles was collected by the detector and averaged by surface type.

Data analysis

Statistical analysis was conducted using the R software (version 4.1.1). Normal distribution of the data groups was performed using the Kolmogorov–Smirnov normality test with ks.test function, and homogeneity of variances was checked using bartlett.test function. Statistical significance was tested using analysis of variance (one-way ANOVA) test. The results were visualized and schematized using ggplot2 library in R and SigmaPlot software (version 12.5).

Results and Discussion

Experimental design verification

We found that the sublimating gas from the dry ice did not leak through the seams or door of the chamber (Fig. 4). The air moved out of the wind tunnel by the horizontal flow generated by the main fan and was circulated to the entire chamber by the mixing fans. Air outflow was only found in the bottom hole of the chamber, which was connected to the detector. This indicates that our system can detect the entire amount of PM resuspended from the surface at the scale of simulating the urban green floor, thus the PM resuspension rate can be quantified.

Figure 4 The process of verifying the experimental chamber design was shown in pictures over time (A–D). The blue arrow indicates the direction of the wind from the main fan on the right. The black arrow indicates the flow in the chamber caused by the main flow. The flow in the chamber was determined by observing the water vapor from dry ice. It was verified that flow was formed only through the outlet at the bottom of the chamber (E), and the door was completely sealed by rubber (F).

Our system has advantages over existing experimental setups in two ways. First, it overcomes the size limitation. A number of existing studies have used a small target surface (usually <100 cm2) because their research objectives focused on the effects of the physical and chemical characteristics of pure materials on resuspension (Kim et al., 2016; Kottapalli & Novosselov, 2021; Li et al., 2022). This small surface area makes it possible to correctly quantify the total number of resuspended particles within the system by counting the number of particles remaining on the surface before and after resuspension. However, to simulate surfaces such as urban green floors, roadsides, and agricultural fields, a larger surface size is required, which makes it impossible to use counting methods to quantify the resuspension rate (Qian & Ferro, 2008; Martuzevicius et al., 2011; Maffia et al., 2021). Our system overcame this limitation because the size was large enough to simulate urban greens, and the closed system was able to capture all the particles resuspended within the system and accurately quantify them.

The second advantage is the versatility of wind tunnels, which enables the use of diverse surfaces. Because of this structure, we can simulate various types of surfaces and conduct replicated experiments with less preparation work, such as cleaning and stabilizing. This versatility of the system implies that it can be used to investigate any interactions between the atmosphere and soil ecosystem.

In summary, the system we developed was large enough to simulate the urban green floor composed of soil and diverse leaves, and cost-effective but able to accurately quantify the total amount of PM resuspension within the system.

Surface roughness and PM resuspension

The Cr index was significantly different among different surface types (Fig. 5, p < 0.001). It was highest on the broadleaf surface, with an average of 10.69 (± 0.41), followed by the conifer and sand surfaces. The results of the surface roughness analysis using Image J software were consistent with those from the roller chain method. The Pearson correlation coefficient between the two was 0.87 (p < 0.005). These results indicate that the broadleaves created a much more uneven floor than the conifers and sand did at the millimeter scale.

The resuspension rates were 14.45 ×10−4 s−1 (± 4.97 ×10−4), 5.35 ×10−4 s−1 (± 4.04 ×10−4), 7.34 ×10−4 s−1 (± 3.31 ×10−4), on broadleaf, conifer, and sand surfaces, respectively (Fig. 6). Our resuspension rates were within the range of the other PM resuspension experiments conducted using various methods, which implies that our design was robust enough to produce reliable results (Krauter & Biermann, 2007; Kim et al., 2016). The resuspension rate of the broadleaf surface was significantly higher than that of the other two surfaces (p < 0.05). These results are inconsistent with our first hypothesis, as well as other existing studies that reported a negative correlation between surface roughness and PM resuspension on a micro-scale (Kearns & Bärlocher, 2008; Liang et al., 2016; Zhang et al., 2020; Zhang et al., 2022). Leaf hairs, which make the surface uneven, were reported to increase the surface area to intercept PM and make it harder for PM to get off the leaves (Zheng & Li, 2019). Moreover, there was a greater chance for PM retention on the leaf surface when the size of the PM was similar to the surface roughness (Nasr et al., 2019). This inconsistency well represented the importance of roughness scales. Our result indicated that high surface roughness at the litter layer scale could rather increase PM resuspension, the opposite trend with the microscopic scale.

The effects of whether a single leaf or layer of leaves on resuspension was also different when we compared our results with those from Kim et al. (2022), which supported our second hypothesis. Kim et al. (2022) reported that a single leaf of broadleaved species with a larger specific leaf area (SLA) or length of margin per leaf area (ML) captured more PM than coniferous leaves. This is contrasted with our result that the layer of broadleaves had a higher resuspension rate. This might be due to the higher turbulence in the array of broadleaves, where erratic flutter and a rugged boundary layer were created. This could not be observed in the experiment using a single leaf (Nicholson, 1993; Henry & Minier, 2014; Zhang et al., 2020).

The contrasting pattern of the effect of surface roughness on resuspension could be related to the turbulence energy varying on different scales. This was explained by the concept of roughness density (Sigal & Danberg, 1990; MacDonald et al., 2016). Roughness density was defined as the distance-to-height ratio in the transverse plane of the target surface. Turbulent energy decreased with increasing roughness density up to a threshold value, beyond which it showed the opposite trend. Following this idea of roughness density and turbulence energy, we suggested a possible conceptual diagram about the relationship between the PM resuspension rate and the surface roughness density by its scale (Fig. 7). On the micrometer scale, negative correlation between PM resuspension and surface roughness was found from the results of previous studies (Li et al., 2022; Zheng & Li, 2019). On the millimeter scale, our data showed that there was a positive relationship between resuspension rate and surface roughness density (r = 0.4654, p < 0.001). The roughness density of the broadleaf surface (3.82 ± 0.20) was greater than that of the coniferous surface (1.30 ± 0.08) and both values exceed the threshold value (0.18) suggested by MacDonald et al. (2016). This indicates that the surface roughness index must include roughness density information to predict PM resuspension at any scale.

Figure 5 Surface roughness represented by Cr index by surface type (p < 0.001).

Figure 6 Total particulate matter resuspension rate by surface type (p < 0.05).

Figure 7 Relationship between surface roughness density and PM resuspension rate.

The solid line on the right shows a positive correlation on the millimeter scale (current study). The dashed line on the left shows a negative correlation on the micrometer scale (existing studies). Correlation coefficient (r) and equation are described in the upper-right.

The change of resuspended PM size distribution

The number of resuspended particles showed trends similar to the total resuspension rates (Fig. 8). The resuspended PM number was highest on the broadleaf layer in less than 0.5 µm and lowest on the coniferous layer through all size ranges. The initial particles have the highest number in particle diameter of 0.4 µm.

Figure 8 The number of resuspended PM.

(A) PM1, (B) PM1–10. The yellow boxes emphasize the appearance of difference between resuspended PM and initial PM sample.

The size distribution of initial particles showed that most of the particles were smaller than 1 µm. Interestingly, the size distribution of resuspended PM had a new peak in the range of 1.5–3 µm, which was not observed in the initial input sample. The result of peak formation in the range of 1.5–3 µm on all the surfaces suggests that the small particles gradually aggregated, without permanently maintaining their original particle size. Our argument was supported by several studies (Fan et al., 2019; Yin et al., 2020). Fan et al. (2019) simulated the PM size enlargement by heterogeneous condensation using numerical model and found the PM smaller than 0.2 µm was condensed to reach at 5 µm. Similarly, Yin et al. (2020) observed particle coagulation on a single leaf surface and reported that the average size of particles grew from 0.48 µm to 3.40 µm. The factors influencing particle coagulation were suggested as initial particle, humidity, wind, temperature and residence time (Fan et al., 2019; Kim et al., 2016; Wu, Davidson & Russell, 1992; Yin et al., 2020). Physiological activity such as transpiration was also reported to affect the coagulation of particles by thermal diffusion (Ryu et al., 2019; Yin et al., 2020). In our experiment, we speculated that wind collision was the most probable factor for coagulation because the constant wind at 4 m/s was blown throughout the experiment which must have generated turbulent flow (Herner et al., 2006). The effect of humidity on coagulation also explained our results because we maintained the relative humidity as relatively high (58 –63%). As we used the dried leaves, the effect of leaves’ physiological activity might have been minimal. Our data on the size redistribution after resuspension indicated that the mean size of resuspended particles was larger than the originally deposited particles. Our size distribution data implied the role of urban green floors in enlarging the particle sizes, which can make PM easier to re-deposit.

Conclusions

This study simulated the resuspension of PM from urban green floors by developing a new experimental design. Our newly developed chamber is cost-effective, accurate, and versatile for quantifying PM resuspension from diverse simulated urban green floors, overcoming the limitations of the existing resuspension experimental setup. The usability of this chamber is not limited to resuspension experiments and can be expanded to experiments investigating the interaction between the atmosphere and any surface with diverse materials.

Our results showed the significantly lower resuspension rate on the coniferous surface than on the other surfaces. This indicated the importance of the selection of plant species when designing the urban green areas for the PM mitigation. The fact that coniferous trees, which are known to be effective in capturing PM, were also good at preventing resuspension, supports the current management practice of planting more coniferous trees in urban greens.

As the effect of surface roughness on PM resuspension is significant both on the micro- and millimeter scales, the roughness scale should be noted when interpreting the results. We adopted the concept of roughness density for explaining the different effects of roughness on multi-scales. The roughness density had a positive correlation with the resuspension rate (p < 0.001) on the millimeter scale. Our results emphasized the need for PM resuspension study under the condition closer to the real situation. The size distribution of resuspended particles showed a new and larger peak around 1.5–3 µm, suggesting coagulation which makes the role of urban green floor even more promising in terms of PM mitigation.

The relationship between roughness density and resuspension rate in our study could be used as input parameters for atmospheric models for simulating PM dynamics. In addition, our implications could provide an insight when interpreting continuous PM monitoring time series data. By diversifying the surface materials and air particulate pollutants, our experimental setup would provide affluent data for model improvement and help identify the important factors influencing PM resuspension.

Supplemental Information

Supplemental Information 1 The concentration of resuspended particulate matter concentration

Click here for additional data file.

Supplemental Information 2 The number of resuspended particulate matter

Click here for additional data file.

Supplemental Information 3 The results of preliminary experiment

The wind speed conditions varied by 1, 2, 4 m s-1. (A) The amount of background suspension, (B) resuspension rates at 4 m s-1 wind speed. The background suspension amount and resuspension rates were different only at 4 m s-1.

Click here for additional data file.

We appreciate Minseop Jeong, Wanseop Jeong and Seok-yeong Jeong, who sincerely helped with the experimental procedures.

Additional Information and Declarations

Competing Interests

Author Contributions

Data Availability

The authors declare there are no competing interests.

Inhye Seo conceived and designed the experiments, performed the experiments, analyzed the data, prepared figures and/or tables, authored or reviewed drafts of the article, and approved the final draft.

Chan Ryul Park conceived and designed the experiments, authored or reviewed drafts of the article, and approved the final draft.

Gayoung Yoo analyzed the data, prepared figures and/or tables, authored or reviewed drafts of the article, and approved the final draft.

The following information was supplied regarding data availability:

The raw measurements are available in the Supplementary File.

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
