# Peer review of "Particulate matter resuspension from simulated urban green floors using a wind tunnel-mounted closed chamber"

_PeerJ, doi:10.7717/peerj.14674_

## Round 0.1 · original submission · Major Revisions

Please revise carefully according to the reviewer's comments.

Reviewer 1 ·

Basic reporting

no comment

Experimental design

no comment

Validity of the findings

no comment

Additional comments

The article is well laid out and easy to understand. However, there are some grammatical mistakes in the article which could be improved. The authors have cited sufficient number of articles to introduce the reader to their work.

I do believe that the article, meets the standards of the journal in its current form and is recommending that it be accepted. However, some suggestions that the authors can look into are mentioned below.

Since temperature and humidity controls are already present in the experimental setup, the effect of temperature and humidity in particle resuspension could be looked into.

The consider including an isometric view of the test chamber in Image 1.

line number 130: It would be better if the authors mention the specific plants from which the leaves were collected. This might help in improving the reproducibility of this study.

lines 156 - 157: Meaning is unclear, consider revising

line number 192: Kindly mention the name of plugin used and cite the author of the plugin

line number 258: mention the units of resuspension rates

Section 2.4: It would be good if the authors mention the specific libraries and functions used in R.

Results and discussion

line no 225: was the presence of leaks checked visually or by some sensors? A simple pressurization test would be a better test for leaks.

Reviewer 2 ·

Basic reporting

The article should include sufficient introduction and background to demonstrate how the work fits into the broader field of knowledge. Relevant prior literature should be appropriately referenced.

Experimental design

Research question well defined, relevant & meaningful. It is stated how research fills an identified knowledge gap

Validity of the findings

All underlying data have been provided; they are robust, statistically sound, & controlled

Additional comments

The paper titled " Particulate matter resuspension from simulated urban green floors using a wind tunnel-mounted closed chamber " presents quite an interesting investigation. and the expected result is helpful in further air quality control in urban regions. But the manuscript need further improvement.
the part of abstract need to be reorganized. In the present manuscript, the main findings were described qualitatively in the part of abstract, and the author is suggested to describe the key results with essential quantity.
2. Ln 46. “Studies have observed that various types of urban greenery reduce atmospheric PM concentrations”, there should be more details to introduce the results.
3. Ln 51. There has two words “adsorption and absorption”, please check it!
4. Ln 56. “stomata,”
5. Ln58: what is the PMs? The authors Should be explain the abbreviation in the manuscript.
6. Another concern is the lack of an adequate scientific description about the difference between soils, coniferous, broadleaved roughness in particles resuspension in introduction. So the hypothesis " PM resuspension was reduced from the litter layer to the bare soil surface because of larger surface roughness" is abrupt.
7. Ln 122. There needs more detail to introduce the PM sample. Example, particle size, component, shape.
8. Ln 136. were the leaves dried at 105℃ for more than 6h? if so, whether effect the leaf surface roughness? Because the leaves may lost all the water.
9. Ln 147. In the experiment, how to set the range of wind speed? The minimum speed is 4m/s, which is maximum speed? There needs more detail to introduce how to set the range of wind speed.
10. Ln 255. Should be “p”, please check all manuscript.
11. Interestingly, the size distribution of resuspended PM110 had a new peak of 1.5-3 µm, which was not observed in the initial input sample. Please provide more detail to discuss.
12. In the part of result and discussion, the author did not fully describe relationship between suspension rate and the size of PM, different wind speed and time, roughness. Because the wind speed, size of PM and roughness are both influence the suspension rates.

---

## Round 0.2 · Minor Revisions

Please revised it according the review comments.

Reviewer 2 ·

Basic reporting

Literature references, sufficient field background/context provided

Experimental design

Research question well defined, relevant & meaningful.

Validity of the findings

All underlying data have been provided; they are robust, statistically sound, & controlled.

Additional comments

The authors has improved the paper according the reviewers' suggestions and advices, but there is still some noticeble question to explain. See suggestions below.
Abstract:
Line 26. What is the millimeter scale, there shoule be given to explain “millimeter scale”.
Results and discussion
Line 338. In this sutdy, the author reported that Our data on the size distribution of resuspended particles implied the role of urban green floors in changing the size of deposited particles, which can make it larger thus less harmful and easier to re-deposit. Please clarify
Conculsion
Line 359-362 Did the authors study the influence of PM resuspension and a single leaf microstructure in the real situation in the results and dicussion? I recommend concluding only on the research findings.

---

## Round 0.3 · accepted · Accept

I confirm that the author has addressed all reviewer comments.